# Effect of Media Composition and Oxygen Tension on Cellular Stress Response and Nrf2 Activation in HepG2ARE Cells

**DOI:** 10.3390/antiox14020137

**Published:** 2025-01-24

**Authors:** Rutt Taba, Marie Põlluaed, Karin Tein, Marju Puurand, Tuuli Käämbre, Anton Terasmaa

**Affiliations:** Laboratory of Chemical Biology, National Institute of Chemical Physics and Biophysics, Akadeemia tee 23, 12618 Tallinn, Estonia; marie.polluaed@gmail.com (M.P.); karin.tein@kbfi.com (K.T.); marju.puurand@kbfi.ee (M.P.); tuuli.kaambre@kbfi.ee (T.K.)

**Keywords:** physiological media, physioxia, Nrf2, ferroptosis, GSH, MDA

## Abstract

Cell models play a central role in preclinical research aimed at the mechanism of disease and drug discovery. The outside environment of the cells, including levels of nutrients and oxygen tension, regulates cellular stress response pathways. Routinely used in vitro disease models often overlook cell growth conditions. This study aimed to evaluate the effect of substituting classic cell media (DMEM) with media matching the nutrient composition of human plasma (Plasmax) on cell viability, the activation of nuclear factor erythroid 2-related factor 2 (Nrf2), glutathione (GSH), and malondialdehyde (MDA) levels by different pharmacological inducers of cell stress. The cells were grown at ambient (~19%) and reduced (5%) oxygen levels. The activation of Nrf2 by ferroptosis activators (erastin and RSL3) was dependent on cell media and oxygen tension. The induction of Nrf2 by an inducer of endoplasmic reticulum stress, thapsigargin, was observable only in cells grown in DMEM and at low oxygen tension. GSH and MDA levels were elevated in Plasmax media. Results indicate that stress tolerance and the activation of Nrf2 in the HepG2ARE cell line depend on the growth conditions, including cell media and oxygen. Cell culture conditions should be critically considered when designing in vitro models of diseases involving oxidative stress.

## 1. Introduction

Cell cultures are widely used in preclinical medical research to understand disease mechanisms and discover novel drugs. Cellular biochemical processes in the in vitro setting must be like the in vivo setting to meet this aim. Formulations of the commonly used cell media were developed in the 1950s. The main aim was to replace animal-born blood-based media, which is of limited supply and often with an unknown composition, with a synthetic mix to improve experimental repeatability and flexibility [1,2]. These included a limited (minimal) number of essential nutrients [1,3,4,5,6], and as a result, concentrations of many nutrients in classical cell media deviate significantly from the amount observed in mammalian plasma [2,7]. For example, the glucose concentration in Dulbecco’s Modified Eagle Medium (DMEM) with high glucose is five times higher than that in the blood of a healthy person [8]. Also, concentrations of other nutrients, such as pyruvate, citrate, and various amino acids, show up to a 10-fold difference in cell media compared to plasma, while some metabolites are absent altogether [7,9]. The composition of popular cell media used in biomedical research vastly differs from mammalian interstitial fluid, resulting in the adjustment of cellular metabolism to a point where it is no longer comparable to that in vivo [7,9].

New cell media formulations were recently developed to overcome this discrepancy, in which concentrations of nutrients and microelements match the concentrations found in human plasma [7,9]. Plasmax (CancerTools, London, UK) and human plasma-like medium (HPLM, Gibco, Billings, MT, USA) are commercially available media. Cancer cells grown in regular DMEM show a metabolic profile that is quite different from cancer biopsy samples. However, when grown in Plasmax media, the metabolome of the cancer cell culture was similar to that of cancer biopsies, demonstrating the benefit of using a physiologically relevant media composition for research involving cell cultures [9].

The proliferation rate of cancer cells in culture depends on the presence of a relatively high concentration of pyruvate [7]. As a result, classical cell media formulations often include high levels of this metabolite. However, high pyruvate levels stabilize hypoxia-inducible factor 1α (HIF1 α), thus also inducing hypoxic transcriptional programs in normoxia [9]. This “pseudohypoxia” state profoundly affects cellular energy metabolism and antioxidant responses. Also, abnormally high arginine concentrations in classical media induce alterations in the urea cycle; thus, the reaction catalyzed by argininosuccinate lyase is reversed in cell culture compared to in vivo settings [7,9]. The cultivation of cell cultures in HPLM media solves such discrepancies [9]. Taken together, the culture of cancer cells in the media approximating the composition of human plasma increases the fidelity of the cellular metabolome, redox state, glucose utilization, and other biochemical parameters [2,7,9].

In addition to the composition of cell media, oxygen tension is another parameter often ignored in maintaining cell cultures [10,11,12]. Oxygen tension varies in different organs but is always less than in ambient air. The lung epithelium and retina are exposed to the highest oxygen levels (approximately 9–11%), while in most tissues, the oxygen concentration is around 4–5%, approximately four times less than in ambient air [10,11,12]. Nevertheless, most cell culture experiments are conducted in ambient air conditions, i.e., at supraphysiological oxygen tension [11,13,14]. At the same time, oxygen tension plays a significant role in regulating cellular metabolism. For example, the activity of transcription factor nuclear factor erythroid 2-related factor 2 (Nrf2), a central regulator of cellular antioxidant responses, is dependent on oxygen tension [11,13,15,16,17]. Thus, cells grown at supraphysiological oxygen tension may display a skewed antioxidant response [11]. Altered antioxidant responses and ferroptosis are hallmarks of diverse age-dependent degenerative disorders, including neurodegeneration, diabetic nephropathy, fatty liver disease, and others [18,19,20]. These are often accompanied by lower levels of reduced glutathione (GSH) [18] and increased levels of end products of lipid peroxidation, such as 4-hydroxy-nonenal and malondialdehyde (MDA) [21]. The activation of Nrf2 is proposed as a therapeutic strategy for many noncommunicable and ferroptosis-related diseases. It is successfully used in clinical practice to treat Friedreich ataxia, a genetic neurodegenerative disorder with iron accumulation [15,22,23,24,25,26,27]. Ferroptosis is directly linked with the intramembrane transport of cystine via cystine/glutamate antiporter (system xC^−^) [20]. Concentrations of both metabolites are lower in Plasmax than in DMEM. Therefore, media composition may also regulate ferroptosis-related pathways, leading to undesirable artifacts using in vitro models in ferroptosis-related research.

Most studies focus on the effect of media composition or oxygen tension separately. Thus, this study aimed to evaluate the impact of various cell stressors on cell viability and the activation of Nrf2 in cells grown in different media and at different oxygen tensions. For this purpose, we evaluated cell survival, glutathione levels, intracellular malondialdehyde concentration, and sensitivity to various cellular stressors in hepatic HepG2ARE cell lines, where the expression of luciferase under the antioxidant response element (ARE) promotor is correlated to Nrf2 activation. Cells were grown in regular cell media (DMEM) or a physiologically relevant Plasmax media. Also, oxygen tension was modified to the physiologically relevant 5% (physioxia) and compared to ambient levels (~19% in the conventional CO_2_ incubator). The effect of various inducers of ferroptosis, endoplasmic reticulum (ER) stress, and direct activators of Nrf2 was evaluated in these conditions.

## 2. Materials and Methods

### 2.1. Cell Culture

This study used human hepatic HepG2ARE cells with luciferase under an Nrf2-sensitive ARE promotor (BPS Bioscience, San Diego, CA, USA, Cat# 60513, kindly provided by Dr. Kattri-Liis Eskla, University of Tartu). Two different cell media were used to maintain the cells—DMEM high-glucose media (Corning^®^, Manassas, VA, USA, Cat# 10-013-CV) supplemented with 10% fetal calf serum (FBS, Capricorn, Ebsdorfergrund, Germany Cat# FBS-12A) and penicillin/streptomycin (Capricorn, Cat# PS-B), and Plasmax media (CancerTools, London, UK, Cat# 156371) with 2.5% FBS and penicillin/streptomycin. For the adjustment of oxygen tension (the 5% O_2_ condition), cells were kept in a humidified hermetic incubation chamber (Billups-Rothenberg, Del Mar, CA, USA, Cat#MIC-101) filled with a gas mix (5% O_2_, 5% CO_2_, rest N_2_) under normal pressure, and the gas mix was changed daily. Part of the cells were grown in a conventional 5% CO_2_ incubator (the 19% O_2_ condition). Cells were adapted to different growing conditions over two weeks before further evaluation.

The following cellular stress inducers and Nrf2 activators were used in the study—(1) H_2_O_2_ (AppliChem, Darmstadt, Germany, Cat# A1134,0250)—1 µM to 4000 µM for MTT and Nrf2 measurement, 2000 µM for GSH and MDA evaluation; (2) RSL3 (MedChemExpress, Monmouth Junction, NJ, USA, Cat# HY-100218A)—0.02 µM to 10 µM for viability and Nrf2 activation, 5 µM for GSH and MDA; (3) erastin (MedChemExpress, Cat# HY-15763)—0.02 µM to 10 µM for MTT and Nrf2, 1.25 µM for GSH and MDA measurement; (4) thapsigargin (MedChemExpress, Cat# HY-13433)—0.078 µM to 5 µM to measure viability and Nrf2 activation, 5 µM for GSH and MDA evaluation; (5) dimethyl fumarate (DMF, MedChemExpress, Cat# HY-17363)—0.39 µM to 200 µM for MTT and Nrf2 activation, 200 µM to measure GSH and MDA levels; (6) sulforaphane (MedChemExpress, Cat# HY-13755)—0.39 µM to 200 µM for viability and Nrf2, 50 µM for GSH and MDA measurement; and (7) bardoxolone (RTA-402 (MedChemExpress, Cat# HY-13324))—0.004 µM to 4 µM for MTT and Nrf2 activation, 0.5 µM for GSH and MDA evaluation.

### 2.2. Nrf2 Activation Assay

HepG2ARE cells were cultivated in 96-well plates suitable for luminescence measurements (Greiner, Kremsmünster, Austria, Cat# 655098) at 40,000 cells/well in 100 μL of media. Cells were allowed to attach for two hours, and after that, compounds were added to the cells and incubated for 24 h. According to the manufacturer’s instructions, cell media was removed, and 100 μL of D-luciferin solution (150 μL/mL; PerkinElmer, Springfield, IL, USA, Cat# 122799) prepared in respective cell media was added to the cells. Cells were incubated for 10 min at 37 °C, and bioluminescence was measured using a FluOStarOmega plate reader (BMG LABTECH, Ortenberg, Germany).

### 2.3. MTT Assay

Cells were seeded to 96-well plates (TPP, Trasadingen, Switzerland, Cat# 92196) at a density of 40,000 cells per well and allowed to adhere for two hours. Compounds were added to the cells and incubated for 24 h in 100 μL of media, after which 10 μL of thiazolyl blue tetrazolium bromide (MTT, Thermo Scientific, Carlsbad, CA, USA, Cat# L11939.06) solution (5 mg/mL in Dulbeccos’s phosphate buffered saline (DPBS, Thermo Scientific, Cat# 158990050) was added to each well, and cells were incubated at 37 °C for 3 h. A total of 150 μL of dimethyl sulfoxide (DMSO, Corning^®^, Manassas, VA, USA, Cat# 25-950-CQC) was added to each well, and cells were incubated for 30 min at 37 °C. Optical density was read at 570 nm and 630 nm using a FluOStarOmega plate reader.

### 2.4. GSH and GSSG Assay

GSH and glutathione disulfide (GSSG) assays were performed using the enzymatic recycling method described by Rahman et al. [28]. Cells were cultured in 8 ml of media with or without compounds on 100 mm cell culture dishes (Greiner, Cat#664160) for 24 h. All standards and reagent solutions were prepared in 0.1 M phosphate buffer (pH 7.5) containing 5 mM EDTA disodium salt (AppliChem, Cat# A2937), called KPE buffer. Cytosolic cell extracts were prepared by resuspending collected cells in an extraction buffer containing 0.1% Triton X-100 (Sigma-Aldrich, St. Louis, MO, USA, Cat# T-9284) and 0.6% sulfosalicylic acid (Sigma-Aldrich, Cat# S2130) in KPE buffer. The cell count was recorded for later analysis. The extracts were stored at −80 °C.

For the GSSG measurement from the cell extracts, GSH was derivatized by 2-vinylpyridine (Sigma-Aldrich, Cat# 132292), and excess 2-vinylpyridine was neutralized with triethanolamine (Thermo Scientific, Cat# 421631000). Standards, derivatized or pure samples, were mixed with 5,5′-dithiobis (2-nitrobenzoic acid) (DTNB, Alfa Aesar, Kandel, Germany, Cat# A14331.03) and glutathione reductase (Sigma-Aldrich, Cat# G3664) in 96-well plates. Absorbance was read immediately after adding β-NADPH (AppliChem, Cat# A1395) at 412 nm using the FluOStarOmega plate reader. Measurements were taken every 20 s for 2 min (totaling 7 readings from 0 to 120 s) and repeated twice for each sample. The concentrations of total GSH and GSSG in the samples were calculated using linear regression based on a standard curve generated from GSH (Sigma-Aldrich, Cat# G4251) or GSSG (Thermo Scientific, Cat# 320225000) standards, ranging from 26.4 nM to 0.103 nM. The final total GSH and GSSG concentrations were expressed as nM per 10^6^ cells. Untreated control values were pooled from all of the experiments for analysis.

### 2.5. Cytosolic MDA Measurement

The Colorimetric Thiobarbituric Acid Reactive Substances (TBARS) assay kit (Cayman chemical, Ann Arbor, MI, USA, Cat#700870) was used to measure cytosolic malondialdehyde in cell extracts prepared for the GSH assay. Briefly, samples of MDA standards were mixed with 10% trichloroacetic acid and freshly prepared color reagent (2-thiobarbituric acid dissolved in 20% acetic acid and 0.7 M NaOH). Tubes were placed in a thermomixer at a temperature of 99 °C for one hour, followed by an ice bath. The solutions were transferred to 96-well plates for absorbance measurement at 530 nm using the FluOStarOmega plate reader. Sample MDA values were calculated from the standard curve (ranging from 0.125 μM to 10 μM), and the concentrations were expressed as μM per 10^6^ cells. For analysis, the control values were pooled from all of the experiments.

### 2.6. Data Analysis

Data were analyzed with GraphPad Prism 9 and Statistica 7. A multivariant ANOVA followed by a Tukey post hoc test was used to determine statistical significance. Data are presented as mean ± SEM of three independent experiments; a *p*-value of less than 0.05 was considered statistically significant. The data’s normality and suitability for the ANOVA analysis were evaluated using Kolmogorov–Smirnov test. The analysis did not include measurement results below the respective assay’s detection limit.

## 3. Results

### 3.1. HepG2ARE Cell Growth and Antioxidant Status Depended on Media Composition and Oxygen Tension

Cell growth conditions had a substantial effect on the untreated cells, where cells grown in DMEM media showed significantly higher viability (F_1, 227_ = 116.04; *p* < 0.001) and had increased Nrf2 activity (F_1, 323_ = 88.16; *p* < 0.001) compared to Plasmax (Figure 1A,B). Basal GSH, GSSG, and MDA levels were considerably elevated in Plasmax media. At 19% O_2_, total GSH was higher in cells grown in Plasmax than DMEM (F_1, 193_ = 32.93; *p* < 0.0001; Figure 1C). There was no effect of oxygen on cells in DMEM (F_1, 193_ = 0.61; *p* = 0.43), whereas GSH levels in cells grown in Plasmax media decreased significantly at 5% O_2_ compared to 19% O_2_ (*p* < 0.05). In addition, there was an interaction between the media and oxygen effect on GSH levels (F_1, 193_ = 10.33; *p* < 0.01). GSSG was higher in cells grown in Plasmax (F_1, 180_ = 73.39; *p* < 0.0001; Figure 1D). Also, GSSG tended to be higher in cells grown at physioxia than ambient oxygen pressure (F_1, 180_ = 4.05; *p* = 0.046). Cells grown in Plasmax media had much higher levels of MDA (F_1, 239_ = 29.08; *p* < 0.0001) but were not dependent on oxygen (F_1, 239_ = 1.97; *p* = 0.16) (Figure 1E).

### 3.2. The Effectiveness of Stress-Inducing Compounds on HepG2ARE Cells in Different Growth Conditions

#### 3.2.1. Cell Viability, Nrf2 Activation, and Redox Status of HepG2ARE Cells in Response to Hydrogen Peroxide Were Dependent on Media

Hydrogen peroxide causes oxidative stress and, at elevated levels, leads to cell death. Both media and O_2_ affect HepG2ARE cells’ response to oxidative stress. Cells grown in Plasmax medium at low oxygen (5%) were more sensitive to H_2_O_2_ than cells grown at an ambient oxygen pressure (F_1, 484_ = 9.63; *p* < 0.01; Figure 2A), and there was an interaction between media and oxygen effects (F_1, 484_ = 8.30; *p* < 0.01) on cell viability. Cell survival decreased by 42% in 5% O_2_ and Plasmax medium at 4000 µM H_2_O_2_. Compared to other conditions, Plasmax/5% O_2_ cells were also significantly more sensitive to H_2_O_2_ at 1000 µM (*p* < 0.05) and 2000 µM (*p* < 0.001).

The activation of Nrf2 by H_2_O_2_ was more pronounced in cells that were grown in DMEM when compared to cells grown in Plasmax (F_1, 536_ = 10.5; *p* < 0.01; Figure 2B). At the two highest H_2_O_2_ concentrations (2000 µM and 4000 µM), Nrf2 activity in DMEM increased sharply, while activity began to decline in Plasmax (*p* < 0.001).

H_2_O_2_ treatment (2000 µM) decreased the GSH levels in almost all conditions (F_1, 218_ = 7.49; *p* < 0.01; Figure 2C). At an ambient oxygen pressure, GSH levels declined in Plasmax media compared to DMEM, where the amount of GSH stayed the same (F_1, 218_ = 7.24; *p* < 0.01). However, the amount of GSSG was not considerably affected by H_2_O_2_ treatment (F_1, 207_ = 2.34; *p* = 0.12; Figure 2D). While the amount of GSSG was somewhat lower in cells grown in Plasmax, the decrease was insignificant.

MDA levels were lower in H_2_O_2_-treated cells in all the conditions (F_1, 261_ = 6.45; *p* < 0.05; Figure 2E). The decrease was most notable in Plasmax media.

#### 3.2.2. HepG2ARE Cells’ Response to Glutathione Peroxidase 4 Inhibitor RSL3 Was More Pronounced in 5% Oxygen

RSL3, an inducer of ferroptosis, had a dose-dependent effect on cell viability (Figure 3A). Cells grown at DMEM/19% O_2_ were more resistant to RSL3 toxicity than cells grown in other conditions (F_1, 165_ = 6.84; *p* < 0.01 (media effect), F_1, 165_ = 30.07; *p* < 0.0001 (oxygen effect)). The largest difference in viability was observed at 5 µM RSL3, where cells grown under conventional conditions were more resistant to RSL3 than cells incubated in the physiological medium at 19% and 5% O_2_ (*p* < 0.001). Cell survival in Plasmax at 5 µM RSL3 was at least 24% lower than in DMEM/19% O_2_.

The activation of Nrf2 by RSL3 was strongest in cells grown at 5% O_2_ (F_1, 267_ = 27.31; *p* < 0.0001; Figure 3B). At the two highest concentrations of RSL3 used in this study, Nrf2 activation was stronger in cells grown in DMEM/5% O_2_ (*p* < 0.05 (5 µM), *p* < 0.01 (10 µM)). The Nrf2 activity of cells incubated in Plasmax/5% O_2_ was higher at lower concentrations of RSL3 compared to other conditions but began to decrease at 10 µM RSL3.

In general, the 5 µM RSL3 treatment did not regulate the levels of total GSH (F_1, 212_ = 0.03; *p* = 0.85; Figure 3C), GSSG (F_1, 204_ = 0.69; *p* = 0.41; Figure 3D), and MDA (F_1, 263_ = 0.0005; *p* = 0.98; Figure 3E). Significant increases in response to RSL3 could only be seen in Plasmax/19% O_2_ compared to DMEM/19% O_2_ (*p* < 0.05 for GSH and *p* < 0.001 for GSSG).

#### 3.2.3. The Impact of the xC^−^ System Inhibitor Erastin on HepG2ARE Cells Depended on Both Media and Oxygen

Erastin inhibits the xC^−^ system, leading to ferroptosis. Cells grown at an ambient oxygen pressure were somewhat more resistant to the toxic effects of erastin than cells grown at 5% O_2_ (F_1, 158_ = 7.92; *p* < 0.01; Figure 4A). Media did not affect erastin toxicity (F_1, 158_ = 0.11; *p* < 0.74), and there was no interaction between oxygen and media effects (F_1, 158_ = 1.16; *p* = 0.28).

The activation of Nrf2 by erastin was more pronounced in cells grown at 19% O_2_ in Plasmax media (F_1, 253_ = 7.57; *p* < 0.01 (media effect), F_1, 253_ = 22.07; *p* < 0.0001 (oxygen effect); Figure 4B) compared to DMEM. Moreover, there was an interaction of oxygen and media effects on Nrf2 activation by erastin (F_1, 253_ = 5.39; *p* < 0.05). At an erastin concentration of 1.25 µM, oxygen had a significant effect in the physiological medium, where Nrf2 was activated to a greater extent in cells incubated at 19% O_2_ than at 5% O_2_ (*p* < 0.01). There was also a significant difference between cells grown in Plasmax/19% O_2_ and DMEM/5% O_2_, where Nrf2 was more active in Plasmax at concentrations of 1.25 µM (*p* < 0.01) and 5 µM (*p* < 0.05).

Erastin treatment at 1.25 µM significantly lowered total GSH (F_1, 214_ = 6.67; *p* < 0.05; Figure 4C), GSSG (F_1, 202_ = 7.73; *p* < 0.01; Figure 4D), and MDA levels (F_1, 260_ = 5.18; *p* < 0.05; Figure 4E).

#### 3.2.4. Oxygen Sensitivity of Thapsigargin Effect on Nrf2 Activation and Redox Status in HepG2ARE Cells Was Determined by Media

Thapsigargin activates endoplasmic reticulum (ER) stress by depleting ER calcium stores, which can lead to apoptosis. Cells incubated at 5% O_2_ showed higher viability than cells at 19% O_2_ after thapsigargin treatment (F_1, 147_ = 16.33; *p* < 0.0001; Figure 5A). Different media did not affect thapsigargin toxicity (F_1, 147_ = 0.76; *p* = 0.38).

The activation of Nrf2 by thapsigargin was dose-dependent, displaying the highest effect in cells grown in DMEM/5% O_2_ (Figure 5B). In addition, the Nrf2 activation was dependent on media (F_1, 200_ = 39.98; *p* < 0.0001) and oxygen (F_1, 200_ = 20.85; *p* < 0.0001). An interaction of oxygen and media effects on Nrf2 activation by thapsigargin can be seen (F_1, 200_ = 54.61; *p* < 0.0001).

Levels of total GSH were strongly upregulated by the 5 µM thapsigargin treatment (F_1, 220_ = 61.86; *p* < 0.0001; Figure 5C). The increase in GSH levels after thapsigargin was more pronounced in DMEM compared to Plasmax (*p* < 0.001 at 19% O_2_ and *p* < 0.01 at 5% O_2_). Oxygen had an opposing impact on the thapsigargin effect depending on media. In DMEM, the GSH increase was relatively higher at 5% O_2_ (*p* < 0.001 compared to 19% O_2_), while in Plasmax a more considerable change can be seen at 19% O_2_ (*p* < 0.01 compared to 5% O_2_).

Thapsigargin generally did not regulate GSSG levels within each condition (F_1, 206_ = 1.48; *p* = 0.22; Figure 5D). A decrease in GSSG in response to thapsigargin could only be observed in Plasmax/19% O_2_.

Thapsigargin increased the amount of MDA in Plasmax media, but it was insignificant overall (F_1, 260_ = 2.06; *p* = 0.15; Figure 5E). At an ambient oxygen pressure, cells grown in Plasmax media showed increased MDA levels, while there was no change in the amount of MDA in DMEM (*p* < 0.05).

### 3.3. The Effect of Nrf2 Activators on HepG2ARE Cells in Different Growth Conditions

#### 3.3.1. DMF’s Effect on the Redox Status of HepG2ARE Cells Depended on Both Media and Oxygen

DMF, an activator of Nrf2 that acts via the electrophilic modification of KEAP1 cysteine residue [22], did not measurably affect cell viability in any of the different conditions (Figure 6A). Oxygen did not affect cell viability after treatment with DMF (F_1, 267_ = 1.23; *p* = 0.26), and there was no interaction between oxygen and media effects (F_1, 267_ = 0.01; *p* = 0.90).

The activation of Nrf2 by DMF was most prominent in cells grown at an ambient oxygen pressure (F_1, 357_ = 12.75; *p* < 0.001; Figure 6B). At 200 µM DMF, HepG2ARE cells grown at 19% O_2_ showed a significantly higher Nrf2 activation than cells at 5% O_2_ (*p* < 0.001). This effect was not regulated by media (F_1, 357_ = 0.11; *p* = 0.74), and there was no interaction of oxygen and media effects on Nrf2 activation by DMF (F_1, 357_ = 0.02; *p* = 0.89).

A total of 200 µM DMF strongly upregulated the amount of GSH (F_1, 214_ = 103.53; *p* < 0.0001; Figure 6C). Oxygen had the opposite effect in different media. The increase in GSH in response to DMF was more pronounced at 5% O_2_ for cells grown in DMEM, while in cells grown in Plasmax, it was more significant at 19% O_2_ (*p* < 0.001 for DMEM and *p* < 0.01 for Plasmax).

The DMF treatment did not regulate GSSG levels within each group (F_1, 193_ = 2.40; *p* = 0.12; Figure 6D), except for the Plasmax/19% O_2_ condition, where GSSG levels increased in response to DMF (*p* < 0.05).

DMF increased the amount of MDA (F_1, 254_ = 19.24; *p* < 0.001; Figure 6E), with the highest rise observed under the Plasmax/5% O2 condition (*p* < 0.05).

#### 3.3.2. HepG2ARE Cells’ Response to Sulforaphane Is Most Evident at 5% Oxygen and Physiological Media

Sulforaphane, also an activator of Nrf2 [22], had a dose-dependent effect on cell viability (Figure 7A). Cells incubated at 19% O_2_ were more sensitive to sulforaphane treatment (F_1, 236_ = 290.25; *p* < 0.0001). A substantial variation in cell viability under different O_2_ conditions can be observed at sulforaphane concentrations from 0.78 µM to 6.25 µM (*p* < 0.01). The media did not affect sulforaphane toxicity (F_1, 236_ = 3.22; *p* = 0.07), and no interaction existed between oxygen and media (F_1, 236_ = 0.01; *p* = 0.93).

Sulforaphane had the strongest effect on Nrf2 activation in cells grown in a DMEM/5% O_2_ at 200 µM (*p* < 0.001; Figure 7B). The induction of Nrf2 by sulforaphane was regulated by media (F_1, 392_ = 12.77; *p* < 0.001), but was not dependent on oxygen (F_1, 392_ = 2.53; *p* = 0.11). There was an interaction of oxygen and media effects on Nrf2 activation by sulforaphane (F_1, 392_ = 6.57; *p* = 0.01).

The sulforaphane treatment at 50 µM decreased the levels of total GSH (F_1, 218_ = 7.96; *p* < 0.01; Figure 7C), GSSG (F_1, 206_ = 5.03; *p* < 0.05; Figure 7D), and MDA (F_1, 256_ = 4.76; *p* < 0.05; Figure 7E). The cells’ response to sulforaphane was most evident in physiological media and DMEM/5% O_2_.

#### 3.3.3. The Impact of Bardoxolone on HepG2ARE Cells’ Nrf2 Activation and Modulation of Redox Status Was Revealed in 5% Oxygen or Physiological Media

Bardoxolone, another activator of Nrf2 that acts via the electrophilic modification of KEAP1 cysteine residue [22], had a distinct dose-dependent effect on cell viability, yet it was similar in all conditions (Figure 8A). Oxygen had no effect on cell viability after treatment with bardoxolone (F_1, 375_ = 0.90; *p* = 0.34), and there was no interaction between oxygen and media effects (F_1, 375_ = 0.88; *p* = 0.35).

The activation of Nrf2 by bardoxolone was regulated by media (F_1, 396_ = 8.11; *p* < 0.01; Figure 8B), with distinct peaks of Nrf2 activation at the 0.5 µM bardoxolone treatment (Plasmax media, both 19% O_2_ and 5% O_2_) and at 1 µM for cells in DMEM/5% O_2_ (*p* < 0.001). Bardoxolone’s effect on Nrf2 activity was not dependent on oxygen (F_1, 396_ = 0.94; *p* = 0.33). However, there was an interaction of oxygen and media effects (F_1, 396_ = 15.17; *p* < 0.001).

Levels of total GSH were significantly increased by the 0.5 µM bardoxolone treatment (F_1, 218_ = 1576.03; *p* < 0.0001; Figure 8C); this effect was dependent on oxygen (F_1, 218_ = 322.60; *p* < 0.0001) and media (F_1, 218_ = 106.20; *p* < 0.0001). The increase in GSH levels in response to bardoxolone was considerably higher at 5% O_2_ (*p* < 0.001 compared to 19% O_2_).

GSSG levels were also strongly upregulated by bardoxolone (F_1, 207_ = 358.73; *p* < 0.0001; Figure 8D), with a more notable increase in physiological media (*p* < 0.001 compared to DMEM).

The amount of MDA in response to bardoxolone did not change (F_1, 258_ = 0.03; *p* = 0.87; Figure 8E).

## 4. Discussion

Cell culture experiments are the cornerstone of preclinical medical research, and the importance of in vitro models can only be expected to increase in the future, as animal models are costly, have ethical limitations, and do not offer the same experimental flexibility. Culture conditions critically modulate cell metabolism and antioxidant pathways, which are causative factors of many diseases. Therefore, cell growth conditions must be considered as an essential part of the design of an in vitro disease model.

This work demonstrates the effects of various cell stress-inducing compounds on cell survival, lipid peroxidation, glutathione levels, and the activation of Nrf2 activity in different growth conditions. Although the impact of cell media on cell metabolism and oxygen tension in culture has been evaluated before, this study aimed to assess Nrf2 activation in conditions that include the modification of cell media and oxygen tension.

Cells grown in Plasmax showed lower optical density in the MTT assay, reflecting a slower proliferation of cells grown in Plasmax than in DMEM. Extended cell doubling times in Plasmax were also reported in previous studies [7]. The growth rate of fibroblasts of Lesch–Nyhan disease (LND) in modified Plasmax media strongly depended on the serum concentration, with a doubling time of 10 days at 2.5% serum [29]. In this study, according to the manufacturer’s instructions, we used 2.5% serum in Plasmax media, which may contribute to a slower proliferation rate of HepG2ARE cells in Plasmax. The state and proliferation of the cell are related to its metabolic activity, including energy production. Thus, mitochondria play a central role in the regulation of cell proliferation [30]. Human placental trophoblast stem cells (hTSCs) displayed increased glycolytic activity, mitochondrial respiration, and proliferation when cultured in Plasmax media [31]. Also, four cell lines cultivated in Plasmax for seven days show increased mitochondrial respiration without significant alteration in mitochondrial mass [32,33]. The incubation of Lesch–Nyhan disease fibroblasts in Plasmax media revealed their increased glycolytic capacity and decreased mitochondrial potential [29]. Thus, the cultivation of various cells in Plasmax generally enhances glycolytic activity. However, the effect of Plasmax on mitochondrial respiration is cell-type specific. Hence, the role of mitochondria in regulating the proliferation rate of HepG2ARE cells remains unknown.

In line with slower proliferation, and thus a lower number of cells, the basal activity of Nrf2 in the luciferase assay is somewhat lower in HepG2ARE cells grown in Plasmax than DMEM; these parameters were insensitive to oxygen tension. Total GSH, GSSG, and MDA levels were higher in cells grown in Plasmax than in DMEM. A higher level of MDA suggests a higher level of oxidative stress or higher sensitivity to oxidative stress in cells grown in Plasmax.

Hydrogen peroxide induces oxidative stress in cells, reducing cell viability [34] and increasing Nrf2 levels and activity [35]. In this study, the effect of H_2_O_2_ on cell survival was most notable in the physiological medium (Plasmax), while H_2_O_2_-induced Nrf2 activity was elevated in HepG2ARE cells grown in DMEM. The increased resistance to H_2_O_2_ and the augmented Nrf2 activity of cells grown in DMEM may be related to the components of the medium. For example, high concentrations of L-cysteine in DMEM can increase Nrf2 levels in cells [36], thereby protecting them from oxidative stress. Also, abundant pyruvate in DMEM can neutralize the harmful effects of hydrogen peroxide on cells [37]. Other culture medium components, such as a high glucose concentration, could also affect Nrf2 activation [38].

Erastin, an inhibitor of system xC^−^ transport and an activator of ferroptosis, did not affect cell survival in our experiments. Still, the induction of Nrf2 by erastin depended on media and oxygen tension, with the heightened induction of Nrf2 observed in cells grown in Plasmax at 19% oxygen. Another inducer of ferroptosis, RSL3, works via different mechanisms and is an inhibitor of glutathione peroxidase 4 (GPx4). The activation of Nrf2 by RSL3 was dependent on oxygen tension. These results suggest that the activity of system xC^−^ transport is dependent on media, which, on the one hand, is in agreement with different cystine concentrations in DMEM and Plasmax (201 μM and 65 μM, respectively) [7]. On the other hand, the activity of GPx4 is more dependent on oxygen tension. Reduced levels of glutathione impair the maintenance of cellular redox status and capacity to counter oxidative stress. Maintaining the proper glutathione levels and GSH/GSSG ratio depends on the import of cystine from outside the cells and the activity of glutathione reductase, which depends on NADPH (Figure 9) [39]. Our results suggest that cells’ protection against ferroptosis (triggered by ferroptosis-inducing compounds erastin and RSL3) in cell cultures depends on media and oxygen tension. The effect of the medium on cell metabolism has also been shown previously, where different metabolic pathways were active in HeLa and A549 cells grown in Plasmax or DMEM, depending on the cell type [32,33].

The activation of endoplasmic reticulum stress plays an essential role in many diseases of proteostatic origin. Interestingly, in this study, thapsigargin, an inhibitor of the sarco/endoplasmic reticulum Ca^2+^-ATPase (SERCA) and an inducer of ER stress, did not affect cell viability but only showed a dose-dependent activation of Nrf2-mediated luciferase signal in cells that were grown in DMEM at physioxia. This is in line with the earlier observations where the higher activity of SERCA was observed at physioxia [11,40]. Thus, thapsigargin-induced ER stress may be more pronounced at lower oxygen levels and results in the enhanced activation of Nrf2. Thapsigargin treatment induced an elevation of total GSH level; however, this effect also depended on media and oxygen tension. Therefore, the optimization of media composition together with oxygen tension is crucial for in vitro studies focused on ER stress and intracellular calcium dynamics [40].

Next, we evaluated the effects of direct activators of Nrf2 DMF, sulforaphane, and bardoxolone on cell viability and the activation of Nrf2 in different conditions. Tissue culture conditions did not affect the cell viability profile of Nrf2 inducers, except for sulforaphane, which showed toxicity at lower concentrations in the cells grown at a higher oxygen tension, regardless of the media. As the activation of Nrf2 by sulforaphane at lower concentrations was not different at any condition, it can be suggested that the toxicity of Nrf2 [15] is more pronounced in cells grown at a high oxygen tension. The activation of Nrf2 by DMF was higher in cells grown at a high oxygen tension. In contrast, the dose–response curve of bardoxolone on Nrf2 activation seems to be shifted to the left in cells grown in Plasmax media. Interestingly, the Nrf2 activators used in this study share the same mechanism of action, namely, the electrophilic deactivation of Kelch-like ECH-associated protein 1 (KEAP1) via its cysteine residue [22]. Nevertheless, the sensitivity of their dose-response curve of Nrf2 activation in HepG2ARE cells on cell growth conditions was different. Thus, even compounds with similar mechanisms of action can display different alterations of their pharmacological profile depending on cell growth conditions.

Previous studies have demonstrated that the glucose utilization and bioenergetics profile of four different cancer cell lines are altered upon their cultivation in Plasmax media and at physioxia (5% oxygen) compared to cells grown in DMEM [41]. The results of this study suggest that in addition to the alteration in cellular metabolism, cells cultivated in Plasmax are also more sensitive to compound-induced stress; conversely, cells grown in DMEM are more protected from cellular stress. In another study, a fascinating effect of media and oxygen tension was observed regarding the impact of estradiol, which lowered peroxide production in cells grown in DMEM at ambient oxygen tension. Still, such an effect of estradiol was absent in cells grown in Plasmax [42]. This raises the question of whether cellular effects and pathologies seen in cellular models at non-physiological conditions occur naturally. In contrast, as demonstrated in a study regarding Lesch–Nyhan disease, the cultivation of the patients’ fibroblasts in physiologically relevant modified Plasmax media made it possible to reveal novel cellular alterations in Lesch–Nyhan disease that were previously masked by the use of standard media [29].

Alterations in cellular metabolism, respiration, and antioxidant response are widely acknowledged to be associated with the progression of cancer and neurodegeneration, which are the focus of many in vitro studies. Therefore, using physiologically relevant media and oxygen tension will enhance the validity of preclinical in vitro models of these diseases. At the very least, the choice of media and oxygen tension must be considered during the creation of the disease model, as the stress-inducing capacity of a given pathology may not be sufficient to overcome the higher antioxidant capacity of cells grown in non-physiological media. Alternatively, even worse, it may introduce an artificial alternative mechanism of toxicity of such a pathology. Also, the curative actions in such non-physiological conditions may not be translatable to in vivo situations.

## 5. Conclusions

From the results of this work, it can be concluded that the survival of cells and the antioxidant response depend on both oxygen and the composition of the medium. Our data shows that the composition of the culture medium significantly impacts the antioxidant response, surpassing the effect of oxygen tension. This might benefit the application of cellular models for drug discovery using high throughput screening (HTS), as a change of media is technically much more accessible than the regulation of oxygen tension in an HTS facility. The results of this study can only be considered preliminary, and thorough research comparing metabolomics and transcriptomic profiles of cells under varying in vitro conditions with corresponding tissue from in vivo conditions will provide a definitive validation of the most optimal and cost-effective conditions for in vitro cell models. Our results demonstrate the importance of cell culture conditions when studying the antioxidant response.

## Figures and Tables

**Figure 1 antioxidants-14-00137-f001:**
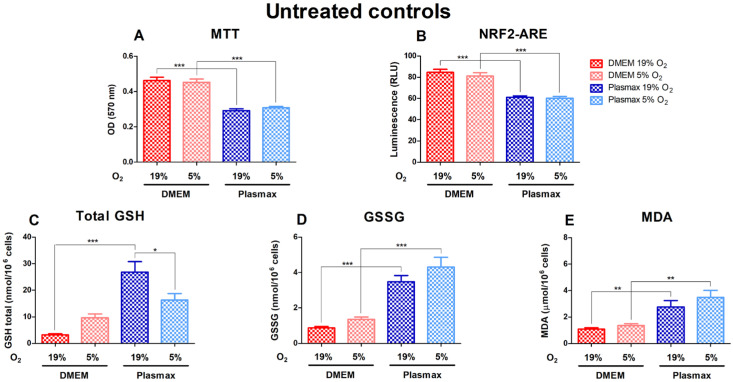
Basal levels of viability (**A**), Nrf2 activation (**B**), glutathione (GSH) (**C**), glutathione disulfide (GSSG) (**D**), and malondialdehyde (MDA) (**E**) measurements in untreated HepG2ARE cells in DMEM and Plasmax media incubated in 19% or 5% O_2_. Data are presented as mean SEM (n = 3). *** *p* < 0.001; ** *p* < 0.01 represent differences between DMEM and Plasmax media. # *p* < 0.05 represents significance between 19% and 5% oxygen. Statistical significance was determined using multiple comparison ANOVA test followed by Tukey post hoc analysis.

**Figure 2 antioxidants-14-00137-f002:**
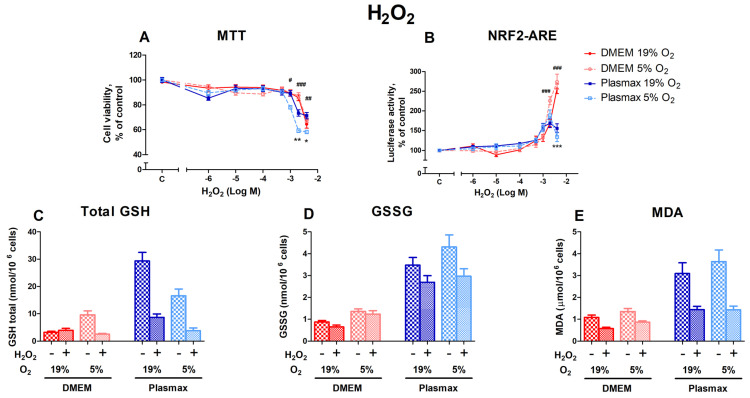
The effect of H_2_O_2_ on HepG2ARE cells is maintained in different media compositions and oxygen concentrations. H_2_O_2_ was applied to the cells at concentrations of 1 µM to 4000 µM for 24 h, and (**A**) cell viability and (**B**) Nrf2 activation were determined. For (**C**) total GSH, (**D**) GSSG, and (**E**) cytosolic MDA measurements, cells were treated with 2000 µM H_2_O_2_ for 24 h. Data are presented as mean SEM (n = 3). *** *p* < 0.001; ** *p* < 0.01; * *p* < 0.05 represent differences between DMEM and Plasmax. ### *p* < 0.001; ## *p* < 0.01; # *p* < 0.05 represent significance between 19% and 5% oxygen. Statistical significance was determined using multiple comparison ANOVA test followed by Tukey post hoc analysis.

**Figure 3 antioxidants-14-00137-f003:**
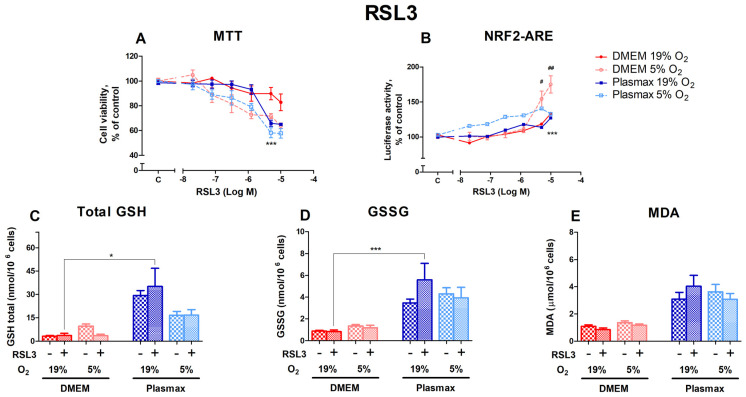
HepG2ARE cells’ response to RSL3 under different media compositions and oxygen concentrations. Cells were treated with 0 02 µM to 10 µM of RSL3 for 24 h, after which (**A**) cell viability and (**B**) Nrf2 activation were determined. (**C**) Total GSH, (**D**) GSSG, and (**E**) cytosolic MDA levels were measured in cells with 5 µM RSL3 for 24 h. Data are presented as mean SEM (n = 3). *** *p* < 0.001; * *p* < 0.05 represent differences between DMEM and Plasmax. ## *p* < 0.01; # *p* < 0.05 represent significance between 19% and 5% oxygen. Statistical significance was determined using multiple comparison ANOVA test followed by Tukey post hoc analysis.

**Figure 4 antioxidants-14-00137-f004:**
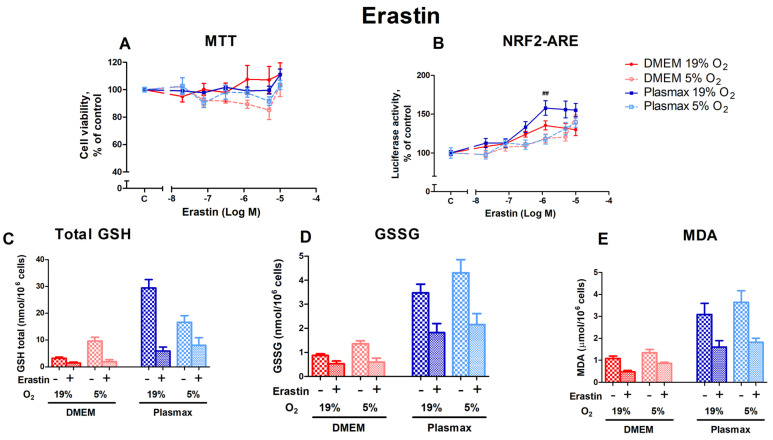
Effect of erastin on HepG2ARE cells grown in different media compositions and oxygen concentrations. A total of 0.02 µM to 10 µM of erastin was added to the cells for 24 h, and (**A**) cell viability and (**B**) Nrf2 activation were measured. For (**C**) total GSH, (**D**) GSSG, and (**E**) cytosolic MDA assessment, cells were treated with 1.25 µM of erastin for 24 h. Data are presented as mean SEM (n = 3). ## *p* < 0.01 represents significance between 19% and 5% oxygen. Statistical significance was determined using multiple comparison ANOVA test followed by Tukey post hoc analysis.

**Figure 5 antioxidants-14-00137-f005:**
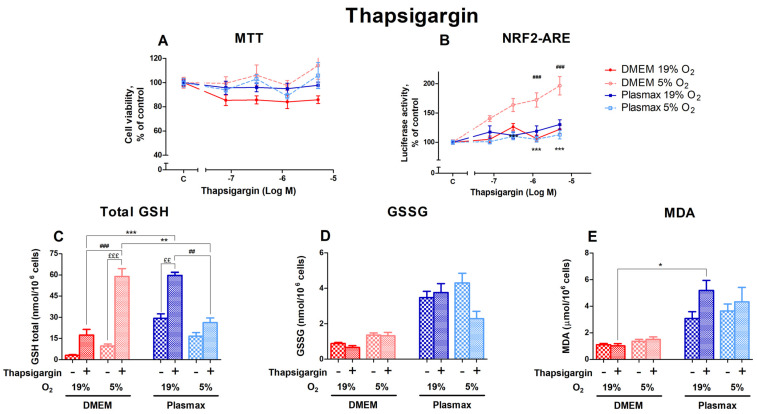
Effect of thapsigargin on HepG2ARE cells grown in different media and oxygen conditions. (**A**) Cell viability and (**B**) Nrf2 activation were measured after 24 h of 0.078 µM to 5 µM thapsigargin exposure. (**C**) Total GSH, (**D**) GSSG, and (**E**) cytosolic MDA levels were measured in cells treated with 5 µM thapsigargin for 24 h. Data are presented as mean SEM from 3 independent experiments per condition. *** *p* < 0.001; ** *p* < 0.01; * *p* < 0.05 for effects between DMEM and Plasmax. ### *p* < 0.001; ## *p* < 0.01 for impact between 19% and 5% oxygen. £££ *p* < 0.001; ££ *p* < 0.01 for stressor effect. Multiple comparison ANOVA test followed by Tukey post hoc analysis was used to determine statistical significance.

**Figure 6 antioxidants-14-00137-f006:**
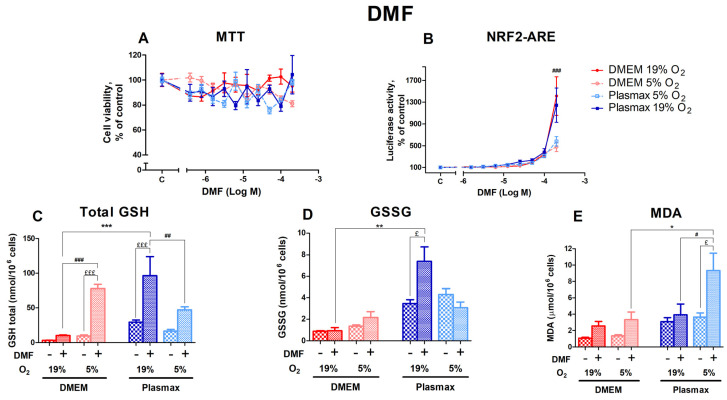
Effect of DMF on HepG2ARE cells grown in different media and oxygen conditions. (**A**) Cell viability and (**B**) Nrf2 activation were measured after 24 h of 0.39 µM to 200 µM DMF exposure. (**C**) Total GSH, (**D**) GSSG, and (**E**) cytosolic MDA levels were measured in cells treated with 200 µM of DMF for 24 h. Data are presented as mean SEM from 3 independent experiments per condition. £££ *p* < 0.001; £ *p* < 0.05 for stressor effect. *** *p* < 0.001; ** *p* < 0.01; * *p* < 0.05 for effects between DMEM and Plasmax. ### *p* < 0.001; ## *p* < 0.01; # *p* < 0.05 for impact between 19% and 5% oxygen. Multiple comparison ANOVA test followed by Tukey post hoc analysis was used to determine statistical significance.

**Figure 7 antioxidants-14-00137-f007:**
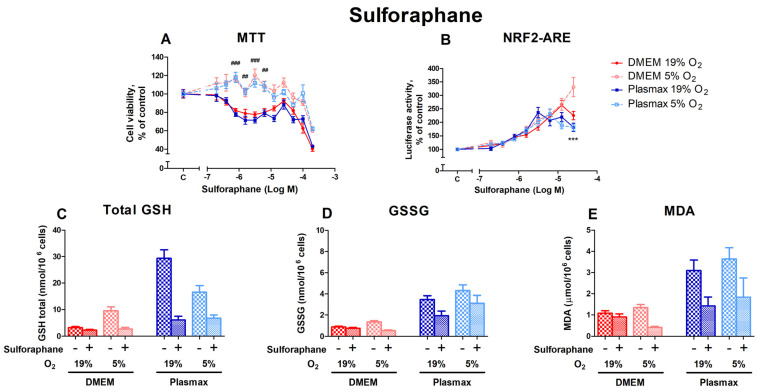
Effect of sulforaphane on HepG2ARE cells grown in different media and oxygen conditions. (**A**) Cell viability and (**B**) Nrf2 activation were measured after 24 h of 0.39 µM to 200 µM sulforaphane incubation. (**C**) Total GSH, (**D**) GSSG, and (**E**) cytosolic MDA levels were measured in cells treated with 50 µM of sulforaphane for 24 h. Data are presented as mean SEM from 3 independent experiments per condition. *** *p* < 0.001 for effect between DMEM and Plasmax. ### *p* < 0.001; ## *p* < 0.001 for impact between 19% and 5% oxygen. Multiple comparison ANOVA test followed by Tukey post hoc analysis was used to determine statistical significance.

**Figure 8 antioxidants-14-00137-f008:**
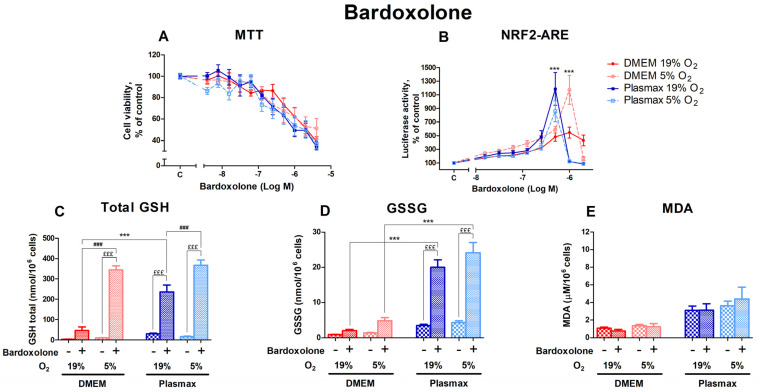
The effect of bardoxolone on HepG2ARE cells is maintained in different media compositions and oxygen concentrations. Cells were treated with 0.004 µM to 4 µM of bardoxolone for 24 h, after which (**A**) cell viability and (**B**) Nrf2 activation were determined. For (**C**) total GSH, (**D**) GSSG, and (**E**) cytosolic MDA measurements, cells were treated with 0.5 µM of bardoxolone for 24 h. Data are presented as mean SEM (n = 3). £££ *p* < 0.001 for Bardoxolone treatment effect. *** *p* < 0.001 represents differences between DMEM and Plasmax. ### *p* < 0.001 represents significance between 19% and 5% oxygen. Statistical significance was determined using multiple comparison ANOVA test followed by Tukey post hoc analysis.

**Figure 9 antioxidants-14-00137-f009:**
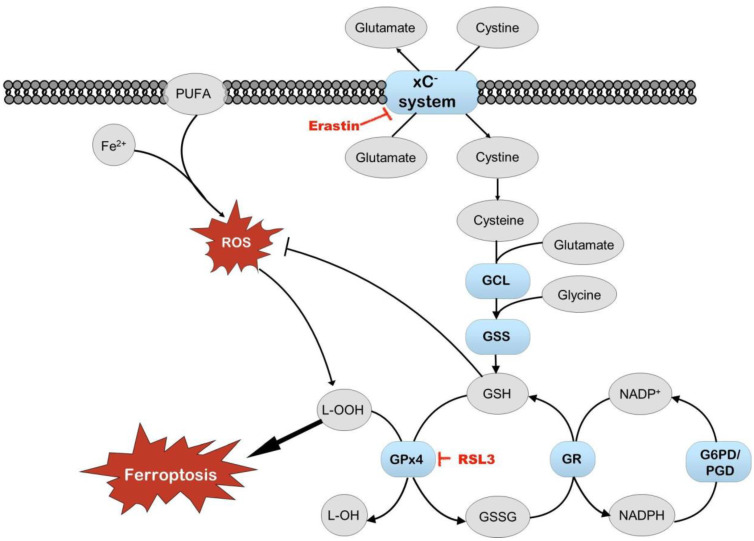
Schematic of ferroptosis. Items in red indicate compounds causing cell stress that can induce ferroptosis. When free iron reacts with polyunsaturated fatty acids (PUFAs) and reactive oxygen species (ROS), it results in lipid peroxidation and may lead to ferroptosis. Nrf2 regulates several enzymes that protect cells from ROS, lipid peroxidation, and ferroptosis (elements shown in blue and bold). Also, glutathione (GSH) protects cells from oxidative stress. Glutathione peroxidase 4 (GPx4) utilizes GSH to reduce lipid peroxides. Oxidized glutathione disulfide (GSSG) is reduced back to GSH by glutathione reductase (GR). The effect of RSL3 on cell viability and Nrf2 activation is sensitive primarily to oxygen tension. The impact of erastin on Nrf2 activation is more dependent on cell media, probably because of the different content of glutamate and cysteine in DMEM and Plasmax. GCL—glutamate cysteine ligase; GSS—glutathione synthetase; G6PD—glucose-6-phosphate dehydrogenase; PGD—phosphogluconate dehydrogenase; NADP/NADPH—nicotinamide adenine dinucleotide phosphate/reduced form of NADP.

## Data Availability

Data are contained within the article and Appendix A. The authors will make the raw data available on request.

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
