# Peer review of "Effect of Media Composition and Oxygen Tension on Cellular Stress Response and Nrf2 Activation in HepG2ARE Cells"

_antioxidants, 2025, doi:10.3390/antiox14020137_

Round 1
Reviewer 1 Report
The work entitled “Media composition and oxygen tension modulate Nrf2 activation in HepG2 cell line” raises an interesting scientific topic, worthy of wider discussion. Therefore, I believe that the topic itself is worthy of publication in the Antioxidants journal. Nevertheless, several issues require improvement or change. First of all, the manuscript is overloaded with data and in some places it is difficult to find your way around. Consideration should be given to improving the description of the results to make them easier to understand. With so many variables, it would be better to always use one term for a given experimental condition. I also suggest the following changes.
1. The title does not fully reflect the content of the work, suggested changing the title to e.g.: “Effect of media composition and oxygen tension on HepG2ARE cells activated by cellular stress inducers” - or something similar. Please note that not only Nrf was changed, the above will be sufficient in case no tendency was demonstrated.
2. Please explain why in the experiments on HepG2ARE cells DMEM with high glucose was used, while these cells are routinely cultured in low glucose and this is also recommended by the supplier of the cells used.
3. Why do the Authors state that the level of MDA, GSH and GSSG in µM per million cells? Wouldn't it be enough to state the concentration in µM? How many cells were seeded and how was the experiment validated?
4. Please provide calibration curves for the performed MDA, GSH and GSSG determinations in the supplementary materials.
5. Please remove lines 184-186.
6. Titles in the results section should be more informative.
7. In the figure descriptions, please add letters labeling to the individual results.
8. Fig. 1A and B, please add the wavelength (A) and unit (B).
9. The data contained in Figures 2-8 A and B are illegible, it is also difficult to see what is significant and what is not, please consider presenting them in a different way.
10. To what extent can the results obtained from three repetitions be considered statistically significant? Was the experiment performed in only 3 repetitions? Were these technical or experimental repetitions? How was the normality of the distribution checked, one of the factors necessary to conduct multifactorial ANOVA analysis? Additionally, in some places there are large differences between groups, e.g. 2.5-fold, and there is no statistical significance (probably due to small size of the group).
11. Did the Authors manage to draw a global conclusion reflected in the obtained results, or perhaps trends indicating a relationship and tendency? Maybe for groups of compounds or dependent on their nature of action? It would be worth including such information in the discussion.
12. Please avoid references to specific figures in the discussion section.
Author Response
Comment : The work entitled “Media composition and oxygen tension modulate Nrf2 activation in HepG2 cell line” raises an interesting scientific topic, worthy of wider discussion. Therefore, I believe that the topic itself is worthy of publication in the Antioxidants journal. Nevertheless, several issues require improvement or change. First of all, the manuscript is overloaded with data and in some places it is difficult to find your way around. Consideration should be given to improving the description of the results to make them easier to understand. With so many variables, it would be better to always use one term for a given experimental condition. I also suggest the following changes.
Response: We are thankful for your input in reviewing this manuscript. Please find the detailed responses below. Where possible, we highlight the revisions and corrections in red in the re-submitted files. In our responses, we indicated any changes in figures by figure number/letter.
We appreciate your comments on improving the readability of our paper. We added several new paragraph breaks to the results section, hoping this would help divide and clarify the data presented. We also rewrote the results concerning GSH, GSSG, and MDA measurements. As requested by the other reviewer, we pooled the controls for GSH, GSSG, and MDA measurements (untreated cell values for each condition from all the experiments). This resulted in changes in Figures 2-8/C-E. We performed a new statistical analysis, which is reflected in the text of the results section.
Detail comments
Comment 1: The title does not fully reflect the content of the work, suggested changing the title to e.g.: “Effect of media composition and oxygen tension on HepG2ARE cells activated by cellular stress inducers” - or something similar. Please note that not only Nrf was changed, the above will be sufficient in case no tendency was demonstrated.
Response 1: We agree that the previous title does not reflect our work completely. We therefore changed the title of our paper to “Effect of media composition and oxygen tension on cellular stress response and Nrf2 activation in HepG2ARE cells.”
Comment 2: Please explain why in the experiments on HepG2ARE cells DMEM with high glucose was used, while these cells are routinely cultured in low glucose and this is also recommended by the supplier of the cells used.
Response 2: We have used this HepG2 cell culture protocol as a basis for our cell culture work with HepG2ARE cells - https://www.encodeproject.org/documents/01ccc7af-f2f3-4d70-bc17-cd82757ef0ca/@@download/attachment/HepG2_protocol.pdf
This protocol used high glucose DMEM, so did we - https://www.cytivalifesciences.com/en/us/shop/cell-culture-and-fermentation/media-and-feeds/classical-media/hyclone-dulbecco's-modified-eagle-medium-dmem-with-high-glucose-liquid-p-05890?srsltid=AfmBOoqwJt6VpgKMQ_ddKaK3BDJQIMHYcTgteVM2hgM5nQQCOCj4G3vt
Comment 3: Why do the Authors state that the level of MDA, GSH and GSSG in µM per million cells? Wouldn't it be enough to state the concentration in µM? How many cells were seeded and how was the experiment validated?
Response 3: HepG2ARE cells were seeded routinely at 2 million cells/100mm cell culture dish at the start of every experiment. As the growth rates vary between DMEM and Plamax media, we counted the cells at the end of the experiments to normalize the subsequent data (GSH, GSSG, and MDA values) for a million cells. We also apologize for a typo in our graphs; the unit should be nmol/106 cells (GSH and GSSG) or µmol/106 cells (MDA); thank you for noticing this. We have changed the units on our graphs (2-8/C-E).
Comment 4: Please provide calibration curves for the performed MDA, GSH and GSSG determinations in the supplementary materials.
Response 4: We uploaded supplementary material with Figure S1. Calibration curve to determine the amount of GSH in HepG2ARE cell extracts; Figure S2. Calibration curve for measuring the amount of GSSG in HepG2ARE cell extracts; and Figure S3. Calibration curve for MDA measurement from HepG2ARE cell extracts. We also included some examples of sample values in addition to the standards.
Comment 5: Please remove lines 184-186.
Response 5: Thank you for your suggestion; we removed the introductory lines in the results section.
Comment 6: Titles in the results section should be more informative.
Response 6: We have added more descriptive titles in the results section, highlighted in red text color.
Comment 7: In the figure descriptions, please add letters labeling to the individual results.
Response 7: Figure 1 was missing labeling letters; we added those to line 211.
Comment 8: Fig. 1A and B, please add the wavelength (A) and unit (B).
Response 8: We apologize for missing that. The OD wavelength and unit for luminescence are added to Figure 1. Thank you.
Comment 9: The data contained in Figures 2-8 A and B are illegible, it is also difficult to see what is significant and what is not; please consider presenting them in a different way.
Response 9: We changed the x-axis, cutting off some of the “empty space” on figures 2-8 A and B. We also increased the font size of the figure legends. We hope it makes the figures more legible.
Comment 10: To what extent can the results obtained from three repetitions be considered statistically significant? Was the experiment performed in only 3 repetitions? Were these technical or experimental repetitions? How was the normality of the distribution checked, one of the factors necessary to conduct multifactorial ANOVA analysis? Additionally, in some places there are large differences between groups, e.g. 2.5-fold, and there is no statistical significance (probably due to small size of the group).
Response 10: We performed all the experiments at least thrice (three experimental repetitions; on different days); plus, each condition in each experiment had three replicates (three technical repetitions). We used this experimental setup as a routine and standard practice in cell culture experiments.
We checked the suitability of the data to perform ANOVA analysis using the Kolmogorov-Smirnov normality test, and our data passed these tests. We also added this info to the Data analysis section in Materials and Methods. We apologize for not including this data in the Materials and Methods section earlier.
We agree that a small sample size might result in no statistical significance in some of the experiments. However, the sample size was large enough to provide several strong statistical significances.
Comment 11: Did the Authors manage to draw a global conclusion reflected in the obtained results, or perhaps trends indicating a relationship and tendency? Maybe for groups of compounds or dependent on their nature of action? It would be worth including such information in the discussion.
Response 11: We came to a global conclusion that media composition impacts antioxidant response more than oxygen. We state that in the Conclusions section, lines 574-577.
Comment 12:. Please avoid references to specific figures in the discussion section.
Response 12: We removed all the references to the results figures from this section. However, we kept referencing Figure 9, which is placed in the discussion section.

Reviewer 2 Report
This manuscript deals with an important issue of experimental setup of in vitro research and how it relates to oxidative stress. Results presented may influence further research directed at in vitro testing, which in turn will be used as a benchmark for in vivo tests.
There are, however, some issues detailed below that need to be resolved by the authors before the manuscript could be accepted.
1. Section Materials and methods does not give explicit numbers for concentrations of substances used in the study. The x-axis gives a rough idea of the spread of concentrations, however, there are different number of concentrations used for different substances, e.g. 6 for erastin, 4 for thapsigargin and 11 for sulforaphane. It would be desirable to describe in detail what exact concentrations were used for each substance, perhaps in a table, to help readers compare with concentrations used throughout literature.
2. In all the subsections of Results an F with lower index is given a value. Please explain where the F comes from and what the indexes mean. Are these the median values for viability percentage and/or GSH, GSSG and MDA concentrations?
3. Figure 1 shows OD for viability and Luminescence for Nrf2-ARE. On the other hand, all remaining figures give cell viability as percentage of control and luciferase activity as percent of control. Please unite Figure 1 A and B panels with the rest.
4. The bar graphs (panels C, D, and E) in each figure show different values for control cells, i.e. the cells marked as (-). Is there any explanation for this phenomenon? I would expect the average (or median) and the error bars to be the same for all control cells. If the controls were performed on different days and different cultures then it should be described in the Materilas and Methods. Or all the control cells results could be pooled making the statistics better (more repeats). Please explain and/or amend.
5. Discussion lines 449 - 456. I would suggest to think about mitochondrial involvement as it may help explain the higher oxidative stress. It could be a nice hypothesis included in the Discussion.
Author Response
Reviewer 2 Comments for Authors
Major comments
Comment : This manuscript deals with an important issue of experimental setup of in vitro research and how it relates to oxidative stress. Results presented may influence further research directed at in vitro testing, which in turn will be used as a benchmark for in vivo tests.
There are, however, some issues detailed below that need to be resolved by the authors before the manuscript could be accepted.
Response: We appreciate your input in reviewing this manuscript. Please find the detailed responses below. Where possible, we highlight the revisions and corrections in red in the re-submitted files. In our responses, we indicated any changes in figures by figure number/letter.
Detail comments
Comment 1: Section Materials and methods does not give explicit numbers for concentrations of substances used in the study. The x-axis gives a rough idea of the spread of concentrations, however, there are different number of concentrations used for different substances, e.g. 6 for erastin, 4 for thapsigargin and 11 for sulforaphane. It would be desirable to describe in detail what exact concentrations were used for each substance, perhaps in a table, to help readers compare with concentrations used throughout literature.
Response 1: Thank you for the suggestion. We did not initially include the concentrations in the methods section because the concentrations for each substance used in this study were stated in the Figure legends. We now have also added the concentrations used to the materials and methods section, lines 118-128.
Comment 2: In all the subsections of Results an F with lower index is given a value. Please explain where the F comes from and what the indexes mean. Are these the median values for viability percentage and/or GSH, GSSG and MDA concentrations?
Response 2: The F values in our results section come from ANOVA statistical analysis, where the F value is the ratio of the between-group variance to the within-group variance. The lower indexes indicate degrees of freedom for treatment and error (FdfTreatment,dfError). In simple terms – a larger F-statistic indicates greater variation between samples than within the samples.
- Comment 3: Figure 1 shows OD for viability and Luminescence for Nrf2-ARE. On the other hand, all remaining figures give cell viability as percentage of control and luciferase activity as percent of control. Please unite Figure 1 A and B panels with the rest.
Response 3: Figure 1 shows the basic levels of viability (panel A) and Nrf2 activation levels (panel B) in untreated cells. We wanted to present non-normalized data here to show the effect of growth conditions on HepG2ARE cells (e.g., viability in DMEM is higher than in Plasmax). Figures 2-8 A and B compare the impact of used substances on viability and Nfr2 activity in different conditions. Therefore, we normalized our data to the controls (untreated cells’ values in each of the 4 conditions) to compare the data better.
- Comment 4: The bar graphs (panels C, D, and E) in each figure show different values for control cells, i.e. the cells marked as (-). Is there any explanation for this phenomenon? I would expect the average (or median) and the error bars to be the same for all control cells. If the controls were performed on different days and different cultures then it should be described in the Materilas and Methods. Or all the control cells results could be pooled making the statistics better (more repeats). Please explain and/or amend.
Response 4: Thank you for noticing the discrepancy. The experiments were performed over several weeks but with the same cultures of HepG2ARE cells, and the different control values are probably a result of this. We pooled all our controls from different days and performed a new statistical analysis. We updated the materials and methods, figures 2-8/C-E, and the results concerning GSH, GSSG, and MDA measurements accordingly (all marked in red text except for figures).
- Comment 5: Discussion lines 449 - 456. I would suggest to think about mitochondrial involvement as it may help explain the higher oxidative stress. It could be a nice hypothesis included in the Discussion.
Response 5: Thank you for the suggestion; we have added references to mitochondrial involvement to the discussion section, lines 453-467.

Round 2
Reviewer 1 Report
I appreciate the contribution of the Authors in improving this manuscript.
The only thing I can suggest is a minor correction of the titles in the results section. The standard definition is: the effect of something...... on something. In the current subtitles we only have the effect of something, without explanation. It is obvious that on the functions of Hepg2ARE cells, but it would be advisable to list it in the individual subtitles of the section.
After these minor corrections, I can recommend the manuscript for publication.
NA
Author Response
Thank you for your feedback. Here are our changes to the titles. We added the line numbers here and marked the new titles in red in the manuscript.
3.1. Cell growth conditions are dependent on media composition and oxygen tension -line 195
HepG2ARE cell growth and antioxidant status depended on media composition and oxygen tension
3.2 The efficacy of stress-inducing compounds in different growth conditions – line 218
The effectiveness of stress-inducing compounds on HepG2ARE cells in different growth conditions
3.2.1 The effect of oxidative stress inducer hydrogen peroxide – line 220
Cell viability, Nrf2 activation, and redox status of HepG2ARE cells in response to hydrogen peroxide were dependent on media
3.2.2 Cells’ response to glutathione peroxidase 4 inhibitor RSL3 – line 256
HepG2ARE cells’ response to glutathione peroxidase 4 inhibitor RSL3 was more pronounced in 5% oxygen
3.2.3 The impact of the xC- system inhibitor erastin – line 285
The impact of the xC- system inhibitor erastin on HepG2ARE cells depended on both media and oxygen
3.2.4 The effect of endoplasmatic reticulum stress inducer thapsigargin – line 313
Oxygen sensitivity of thapsigargin effect on Nrf2 activation and redox status in HepG2ARE cells was determined by media
3.3 The potency of Nrf2 activators in different growth conditions – line 350
The effect of Nrf2 activators on HepG2ARE cells in different growth conditions
3.3.1 Nrf2 activator DMF’s effect on cells – line 353
DMF’s effect on the redox status of HepG2ARE cells depended on both media and oxygen
3.3.2 Cells’ response to Nrf2 activator sulforaphane – line 386
HepG2ARE cells’ response to sulforaphane was most evident at 5% oxygen and physiological media
3.3.3 The impact of Nrf2 activator bardoxolone – line 414
The impact of bardoxolone on HepG2ARE cells’ Nrf2 activation and modulation of redox status was revealed in 5% oxygen or physiological media

Reviewer 2 Report
This is a manuscript focused on the effect of media composition and oxygen partial pressure on signals related to oxidative stress.
I am satisfied with the authors response to my comments. I have no further comments.
Author Response
Thank you for the time and effort in reviewing our manuscript.